# Changing face of socio-economic vulnerability and COVID-19: An analysis of country wealth during the first two years of the pandemic

**Víctor Pérez-Segura**[1]*, **Raquel Caro-Carretero**[2], **Antonio Rua**[3]

**1** University Institute of Studies on Migrations, Comillas Pontifical University, Madrid, Spain, **2** Industrial Organization Department, ICAI-School of Engineering, Comillas Pontifical University, Madrid, Spain, **3** Department of Quantitative Methods, Comillas Pontifical University, Madrid, Spain

* vperezs@comillas.edu

**Data Availability Statement:** All files are available from the Github repository: https://github.com/vicperez/Socioeconomic-Vulnerability-and-COVID-

## Abstract

There are numerous academic studies on the relationship between population wealth and the incidence of COVID-19. However, research developed shows contradictory results on their relationship. In accordance with this question, this work pursues two objectives: on the one hand, to check whether wealth and disease incidence have a unidirectional and stable relationship. And on the other hand, to find out if the country's statistical production capacity is masking the real incidence of the COVID-19 pandemic. In order to achieve this objective, an ecological study has been designed at international level with the countries established as study units. The analytical strategy utilized involves the consecutive application of cross-sectional analysis, specifically employing multivariate linear regression daily throughout the first two years of the pandemic (from 03/14/2020 to 03/28/2022). The application of multiple cross-sectional analysis has shown that country wealth has a dynamic relationship with the incidence of COVID-19. Initially, it appears as a risk factor and, in the long term, as a protective element. In turn, statistical capacity appears as an explanatory variable for the number of published COVID-19 cases and deaths. Therefore, the inadequate statistical production capacity of low income countries may be masking the real incidence of the disease.

## Introduction

Close to four years since the first COVID-19 cases were detected, coronavirus continues to circulate around the world posing a threat to public health. During these four years, an intense scientific production has been generated on the risk factors that catalyze COVID-19 disease. The risk factors identified come from different dimensions, such as environmental [1,2], biological [3,4], political [5] or social. Within this last dimension, several empirical studies have found that territorial wealth is a protective factor against the incidence of COVID-19, in countries such as the United States [6–9], United Kingdom [10,11], Peru [12], Brazil [13], or at the international level [14].

However, there is also research that has found opposite results, even for the same cities. In Brazil, Martins-Filho et al. [15] found that while the most disadvantaged territories reported a

19.-Two-years-of-pandemic-analysis-on-a-global-scale.git.

**Funding:** The authors received no specific funding for this work.

**Competing interests:** The authors have declared that no competing interests exist.

**Abbreviations:** AIDS, Acquired immunodeficiency syndrome; COIVD-19, Coronavirus disease 2019; GDP, Gross domestic product; HC3, Version of covariance matrix consistent with heteroscedasticity; ILO, International Labour Organization; WHO, World Health Organization.

higher rate of deaths associated with COVID-19, the most affluent territories had the highest rate of contagion. Abedi et al [16] arrive at similar results for the seven most affected counties in the United States, where the wealthiest territories sustaining the highest rates of contagion. This lack of convergence in the results may be caused by different reasons like the choice of variables, techniques, or spatio-temporal frames.

Regarding the last issue, throughout the pandemic period the spatial distribution of hot spots has varied over time. Initially, the disease was especially prevalent in the richer countries. This relationship was eventually diluted by the accelerated but delayed increase in cases in less developed countries [17]. Martinez-Alvarez et al [17] suggest that this change is the result of reduced international connectivity in some African countries. This argument could be extended to other impoverished territories with low air traffic where COVID-19 has been slower to manifest itself. Although there are also studies that point out that in some African countries the incidence has been underestimated [18,19].

This is not the first time that a pandemic outbreak has started affecting a group a high socioeconomic status and then moved on to affect a more vulnerable population. In the 1980s, the term "changing face" was coined to denote the process of changing the predominant profile of the AIDS patient from homosexual or bisexual and economically affluent to members of the most disadvantaged strata of society: ethnic minorities, individuals with low socio-economic status and drug addicts [20]. Subsequent studies on AIDS pandemic have found that this process of *changing face* has being occur over time [21].

Based on the concept of the changing face and the evolution of the international distribution of hot spots, we hypothesize that the relationship between wealth and disease is not deterministic. On the contrary, we propose that the relationship between the two is dynamic, showing a different relationship depending on the time frame studied.

## Metodology

Many of the empirical papers that have studied the relationship between social vulnerability and COVID-19 have done so by applying cross-sectional analyses [6–10,12,13,15,16]. However, this type of static approach does not allow the influence of the time dimension on the phenomenon to be captured. To solve this limitation an ecological study has been proposed at country level (N = 216) based on the sequential application of cross-sectional studies. From March 11, 2020, the day on which the WHO declared the outbreak as such (WHO 2020), to April 1, 2022. Specifically, it is applied a multivariate regression model on fatalities and infections for each pandemic day for two years. This strategy allows us to compare the regression coefficients at different times to check whether the covariates maintain the type of relationship with the dependent variables over time.

The regression coefficients have been estimated by ordinary least squares. However, because of the recurrent presence of heteroscedasticity in some model´s residuals, where required, parameters were estimated using the heteroscedasticity-resistant covariance matrix [22]. Specifically, the HC3 matrix is the most recommended option for small samples [23].

The analyses consisted of the application of different regression models, each with a different number of variables (see Eqs 1–3). In the first model, the explanatory capacity, and the association of the country's wealth with deaths and contagions were studied for each day during the period studied. The second model was designed to examine the behavior of the statistical production capacity variable and its relationship with the behavior and explanatory capacity of the wealth variable. The last model includes the explanatory variables considered above, together with a large number of other variables identified in the literature review as risk

factors for COVID-19.

$$V.Dependient_{(cases/deaths)}$$
$$= Intercept + \beta_{1(c/d)} \cdot GDP\ per\ capita + \beta_{3(c/d)} \cdot Stringency\ Index + \beta_{4(c/d)} \cdot Age + \beta_{5(c/d)}$$
$$\cdot Life\ espectancy + \beta_{6(c/d)} \cdot Population + \beta_{7(c/d)} \cdot Temperature + \varepsilon_{(c/d)} \qquad \text{Eq1}$$

$$V.Dependient_{(cases/deaths)}$$
$$= Intercept + \beta_{1(c/d)} \cdot GDP\ per\ capita + \beta_{2(c/d)} \cdot Statitistic\ capacity + \beta_{3(c/d)}$$
$$\cdot Stringency\ Index + \beta_{4(c/d)} \cdot Age + \beta_{5(c/d)} \cdot Life\ espectancy + \beta_{6(c/d)} \cdot Population + \beta_{7(c/d)}$$
$$\cdot Temperature + \varepsilon_{(c/d)} \qquad \text{Eq2}$$

$$V.Dependient_{(cases/deaths)}$$
$$= Intercept_{(c/d)} + \beta_{1(c/d)} \cdot GDP\ per\ capita + \beta_{2(c/d)} \cdot Unemployment + \beta_{3(c/d)}$$
$$\cdot Education\ equality + \beta_{4(c/d)} \cdot Health\ expenditure + \beta_{5(c/d)} \cdot Rural\ population + \beta_{6(c/d)}$$
$$\cdot Statitisticcapacity + \beta_{7(c/d)} \cdot Age + \beta_{8(c/d)} \cdot Life\ espectancy + \beta_{9(c/d)} \cdot Stringency\ Index$$
$$+ \beta_{10(c/d)} \cdot Population + \beta_{11(c/d)} \cdot Temperature + \varepsilon_{(c/d)} \qquad \text{Eq3}$$

## 2.2 Measures

Since this is an ecological study at the international level the units of analysis selected are countries (N = 216). The data for the variables used were retrieved from different open data sources: Climate Change Knoweledge Portal, Data World Bank, Our World In Data and V-dem. Since this is a macro analysis at the national level on data from secondary data sources, the work does not jeopardize the privacy of the participants. All data used in this analysis are publicly available and can be found at: https://github.com/vicperez/Changing-Face-of-Socio-Economic-Vulnerability-and-COVID-19.

**2.2.1. Dependent variables.** *Total cases*: Total confirmed COVID-19 cases [24]. It assumes the dependent variable of the regression models on infections. This variable comprises the time series of cases of infections detected per day. Each regression model was adjusted for the value of this variable corresponding to each day.

*Total deaths*: Total deaths attributed to COVID-19 [24]. The regression models in this study utilize the dependent variable of total number of deaths, representing the cumulative count of recorded deaths. Each regression model is adjusted based on the corresponding cumulative value of this variable.

The absolute measure has been chosen for both indicators instead of the relative measure because the the interest of the research is to know the magnitude of cases (absolut number), not its level of affection (proportion or rate). Furthemore relative measure inflates the incidence in countries with smaller populations. An example of this is that of the five most affected countries in relative terms of contagion on 11/29/2021 only two exceed one million inhabitants, Slovakia with 5.45 million inhabitants and Georgia with 3.71 million [24]. This reflects the given bias of the relative measure in representing the level of outbreak severity.

**2.2.2 Independent variables.** *GDP per capita*: Value of gross domestic product per capita expressed in current international dollars and converted by purchasing power parity [25]. It was selected because it is a summary measure of the economic level of the overall country. This variable was significant as a predictor of the incidence of the 1918 flu pandemic [26].

*Statistical capacity*: Statistical capacity measures a country's ability to collect and produce accurate pandemic statistics, as well as transparency in their publication. This variable has been estimated through the Odin (Open Data Inventory) index. Odin Index measures how complete a country's statistical offerings are and whether its data meet international standards of openness. The variable is a standardized measure whose scores range from 0 to 100, where 100 is the highest degree of statistical quality [27].

*Education Equality Index*: Indicator on the level of attainment of a basic education among the population that allows the exercise of the individual's basic rights. The indicator is a five-level scale (0–4). Where zero is extreme poverty (75% of children receive low quality education), one is unequal (at least 25% of children receive quality education), somewhat equal (10 to 25% receive low quality education), three is relatively equal (only 5% to 10% receive low quality education) and 4 equality (where low education only affects less than 5%) [28]. Educational level has been shown to be a significant element in different studies, where it has shown a paradoxical behavior, being positively associated with deaths [12] and negatively associated with infections [16].

*Unemployment rate*: Percentage of the unemployed population in relation to the total labor force. The indicator is an ILO modeled estimate, which ensures comparability between countries [25]. Indicator selected for its informative quality on the country's level of development.

*Health care expenditure per capita*: Health care expenditure per capita expressed in the current international dollar converted by purchasing power parity [25]. Health spending is used as a proxy for health quality, which is considered a key factor in combating the pandemic.

*Rural popultion rate*: Indicator on the prevalence of rural population in the country, Estimated from the difference between the total population and the urban population [25]. Rutter et al. [29] found a higher incidence of deaths in the H1N1 influenza pandemic in urban territories than rural areas.

*2.2.3.1 Control variables.* Taking into account the large number of variables with potential influence on the spread of the disease, it was decided to introduce a series of control variables, in order to manage the spurious relationships of the results. Taking into account the criticism of Firebaugh [30] on the indiscriminate use of control variables only those variables with sufficient theoretical support in addition to a sufficient level of association ($|r| > 0.1$) have been introduced, as indicated by Becker et al [31]. To these internal reality checks we added a robustness test that consisted of replicating the analyses for all possible combinations in the introduction of control variables. The robustness checks support the consistency of the results presented.

*Age*: Average age of the country's population [24]. Age has been shown to be a risk factor in relation to disease severity [32]. Its use is frequent in different research works as a control variable [12,33].

*Life expectancy*: Life expectancy at birth [24]. Given the existence of proven health variables, such as obesity, diabetes, hypertension with influence on the development of the disease [34], this variable has been included as an indicator of the general health status of the country's population.

*Stringency Index*: Indicator of the stringency of containment measures during the pandemic. Indicator from 0 to 100, where 100 is the highest possible level of stringency of containment policies [35]. The type of stringency has proven to be a key element on the pandemic spread of COVID-19 being an obligatory control variable [36].

*Population*: number of inhabitants of the country [25]. Since we have used absolute and not relative measures as dependent variables, it has been considered necessary to include the number of inhabitants of each country as a control variable to control the effect of the different population sizes of the countries on the results.

*Temperature*: Average monthly temperature constructed from the average value between the years 1991–2020 [37]. Several studies have proven that temperature is a risk factor for the spread of COVID-19 [1].

## Results

Initially, a basic linear regression model was applied to both: cases and fatalities. This model just includes the explanatory variable GDP per capita, together with the control variables (see Eq 1). The results of this model (Fig 1A) showed that GDP per capita appears as a significant predictor of infections during the first 147 days of the pandemic, maintaining a positive relationship with cases. This implies that during this period the richest countries were the most affected by the disease at contagion level. In the model on deaths (Fig 1B) GDP per capita only appears as significant in a late phase of the pandemic, maintaining a negative association. When the wealth of the country appears as a protective factor against long-term COVID-19 mortality.

Subsequently, a second linear regression model was applied (see Eq 2), where, statistical production capacity is added to the previous one as confusión variable. In the set of models carried out on contagions the new variable shows (Fig 2B) a high degree of significative consistency, in a positive and relatively stable manner, showing a slight upward trend. Furthemore statistical capacity produces a notable effect on the GDP per capita variable (Fig 2A). On the one hand, it reduces the number of significant days of the variable as a predictor at the beginning of the pandemic, but on the other hand, it improves its significance at the end of the pandemic. At which point the relationship is reversed, with a negative association between contagion and wealth.

Meanwhile, for the set of regression models applied to the dependent variable deaths, the inclusion of the confusion variable statistical production capacity improves the behavior of the GDP per capita variable (Fig 2C) by increasing the number of days in which it appears as a

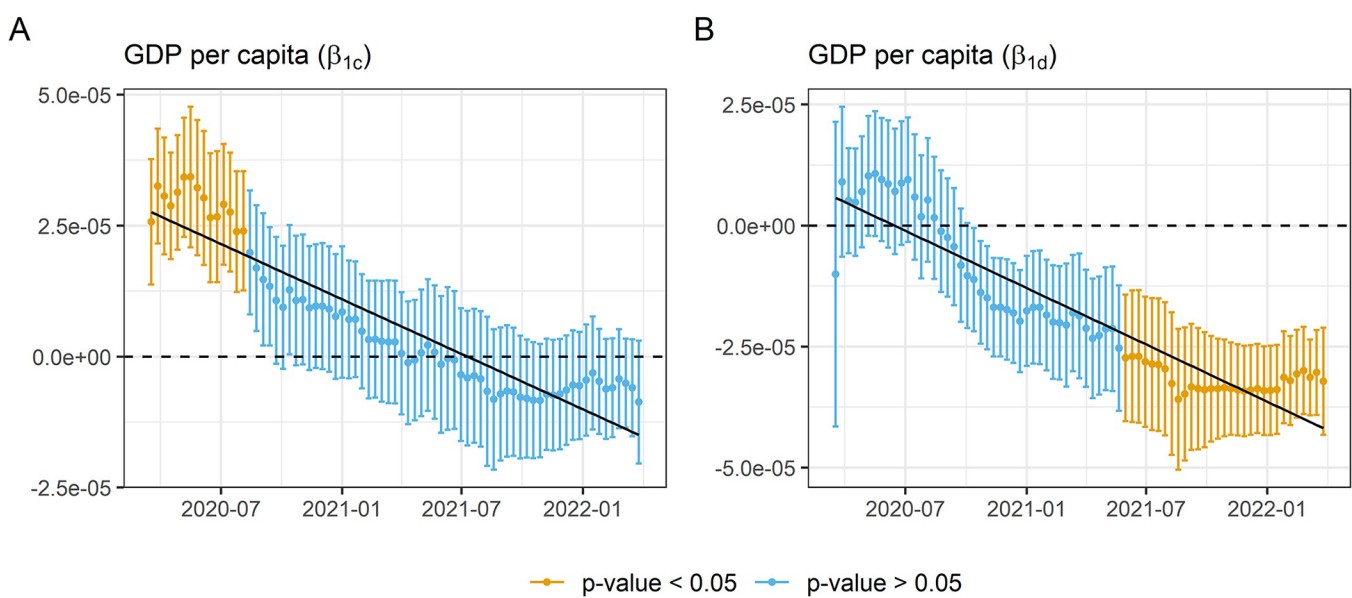

**Fig 1. Evolution of the regression coefficients (GDP + control variables).**

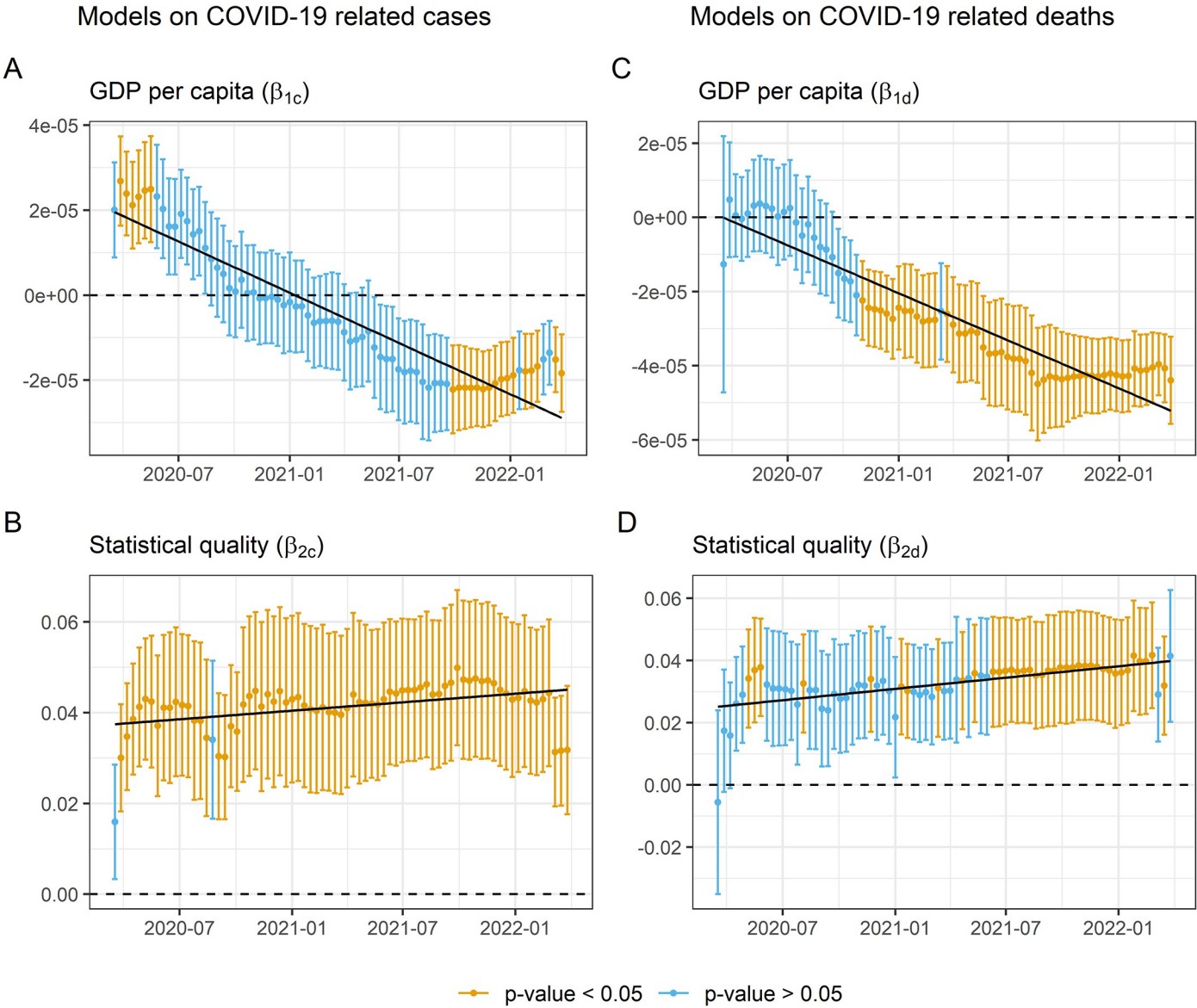

**Fig 2. Regression coefficients evolution (GDP + Statitistic capacity + control variables).**

significant predictor. However, the significance of the statistical capacity variable (Fig 2D) is not as consistent as for the COVID-19 cases, being only significant in 51% of the regressions.

Finally, a third regression model design was performed including the rest of the explanatory variables (see Eq 3). In the set of regressions on COVID-19 cases, the GDP variable (Fig 3A) drops considerably in performance, being barely significant for only 30 days at the end of the period maintaining a negative association. The unemployment rate (Fig 3B), on the other hand, presents better results in terms of significant consistency, as it reflects a relatively stable positive and significant association (it is significant in 39% of the regressions) as of end-September 2020. The educational equality variable (Fig 3C), on the other hand, shows a solid behavior since it is the variable that is significantly associated the most days, practically the entire period except for the first weeks of the pandemic. The association is negative, assuming that the more equality the less cases, and the trend is relatively stable. Although it reveals a

## Models on COVID-19 related cases

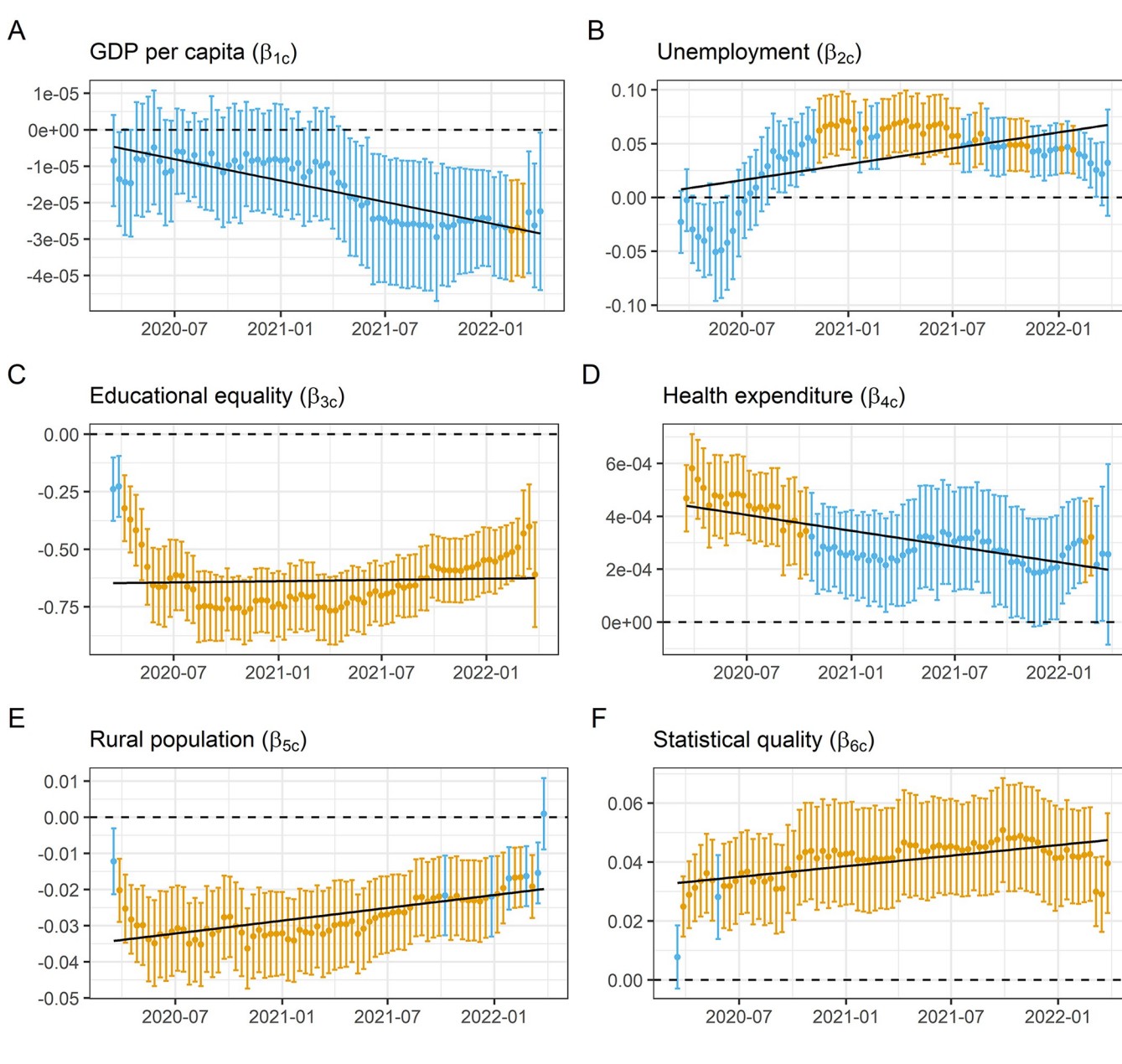

**Fig 3. Evolution of the regression coefficients of the full model on COVID–19 infections.**

sharp decline at the beginning followed by a long period of valley and a slight recovery at the end. The health expenditure variable (Fig 3D) shows a paradoxical behavior, being positively associated with the number of infections during practically the entire year 2020. The rural population rate (Fig 3E), however, shows a more understandable behavior, being negatively associated with the number of infections during the whole period. Finally, the statistical capacity (Fig 3F) retains the effectiveness, reporting a significance rate similar to that reflected in the basic model.

Models on COVID-19 related deaths

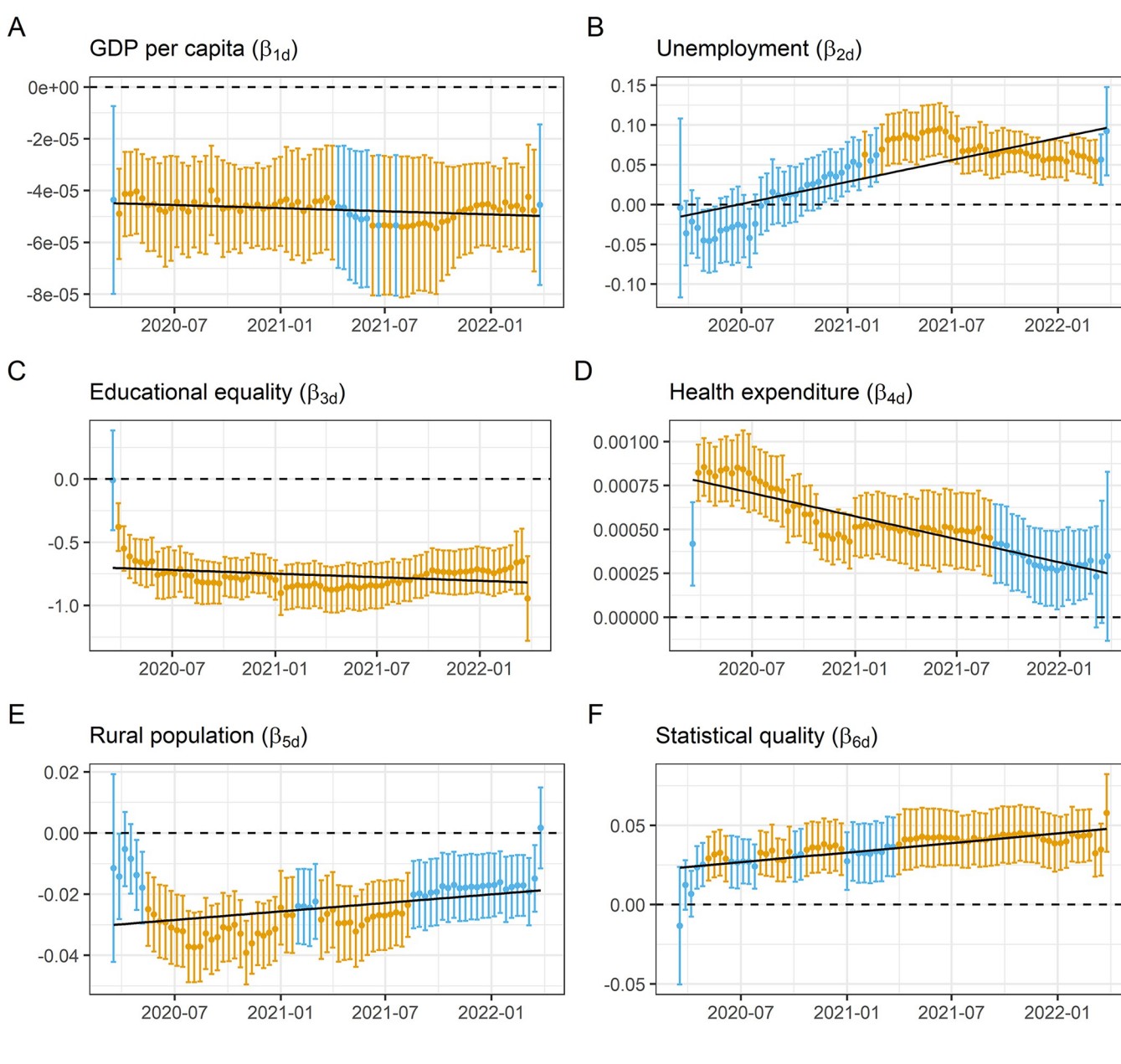

**Fig 4. Evolution of the regression coefficients of the full model on COVID–19 deaths.**

When the complete model is applied to the dependent variable deaths, similar results are obtained at the level of direction and trend of the association, but there are notable differences at the level of significance. The GDP per capita variable (Fig 4A) increases considerably the number of regressions in which it is significant with respect to the basic model, being so during practically the entire pandemic period. Reflecting a stable trend with a negative association. Which means that at the level of deaths, GDP per capita appears as a protective mechanism throughout the pandemic. The unemployment rate (Fig 4B) has a similar result to that

reflected for cases becoming significant late in the pandemic, although this time in a more consistent manner. The educational equality variable (Fig 4C) again has a very consistent explanatory capacity, but this time the final decrease in the degree of its association is less than in the case of contagions. The variable health expenditure (Fig 4D) increases the number of regressions in which it is significant with respect to the regressions on contagions, being so for 535 consecutive days and showing a slight decreasing trend. The proportion of rural population variable (Fig 4D) and the statistical capacity variable (Fig 4F) report very similar behavior to those shown in the batch of models on cases, but in both cases reducing their explanatory capacity.

## Discussion

The objective of the research was to determine the relationship between the country's wealth and the incidence of COVID-19. Under the hypothesis that their relationship is dynamic and changing over time. By successively applying a total of 747 (cross-sectional analysis), we were able to verify significant changes in the association between the variables studied and the incidence of COVID-19 during the first two years of the pandemic.

The first thing that has been noticed is that deaths and infections behave differently with respect to the socioeconomic status of the country. While wealth is associated initially as a risk factor with cases and at the end of the period as a protective element, for deaths it always appears as a protective element. These results coincide with those of Abedi et al [16] where death and infection are associated in the opposite way with the wealth of the territory. Moreover, this dynamic behavior of the country's wealth with contagion gives coherence to the contradictory results of the previous research [6–10,12,13,15,16]. Furtheremore analyses have also shown that different indicators of inequality and economic development, such as educational equality and the unemployment rate, are significantly associated with both contagions and deaths, with the effect of the variable increasing over time in both cases. A striking aspect is that the inclusion of these mechanisms in the regression models meant the loss of significance of wealth as a predictor of contagion, contrary to what happened in the models on deaths, where this variable increased its significance notably with respect to the previous models. Therefore, wealth in itself is not a risk factor for contagion, but rather it is inequality and structural deficiency that have resulted in a differential impact of the pandemic. In summary, the behavior of the variables suggests a process of changing face where the disease has been affecting the most disadvantaged countries over time.

In line with the second objective of this work, the behavior of certain variables informs about a possible underestimation in the accounting of infections and deaths. The statistical capacity variable is very consistent, especially in the case of contagions. The behavior of the variable investment in health shows a behavior that is deduced in the same direction. According to the analyses, the higher the investment, the higher the incidence figure. This result is similar to that found by Bamgboye et al [18], who found the same direction in the association between the Human Development Index and infections and deaths. The interpretation proposed here to explain this paradoxical behavior is that the quality of health services is directly related to the capacity to diagnose and account for infections and deaths. This explanation makes more sense in view of the particularly significant consistency of the health expenditure variable in the models on deaths. In addition to its higher degree of association during the early period of the pandemic, a time of high saturation of health services and low undersanding of the behavior of the disease [38].

A similar interpretation can also be made with respect to the proportion of the rural population. Its behavior shows an expected behavior in accordance with past infectious events,

where the higher the proportion of rural population, the lower the incidence of the disease [29]. This result is plausible given that the rural environment is characterized by a space of lower mobility and population density in relation to the city. However, it can also be deduced that the rural environment is presumably also an area with a worse record of infections and deaths. The lack of administrative infrastructure in isolated and underdeveloped environments may be playing a fundamental role in the real picture of the impact of the catastrophe in these territories.

Therefore, these three variables: health expenditure, rural population rate and statistical capacity are understood to be in some way informing the capacity to identify cases and deaths and their behavior suggests that there has been an underestimation in countries with a lower level of development.

## Conclusion

The methodological strategy of sequentially applying 747 cross-sectional analyses has allowed us to understand the temporal dynamics of the pandemic in relation to socioeconomic risk factors. Each cross-sectional study has served as a still image which, taken together, has made it possible to compose a moving picture.

This approach represents an innovative element by conceiving the association between the country's wealth and the incidence of COVID-19 as a dynamic process. Consideration of the time perspective has made it possible to identify a process of "changing faces", whereby a change in the profile of the vulnerable population takes place. Initially, it was the rich countries that were most exposed to the pandemic in terms of infections, which did not correspond to higher lethality, and later it was the countries with the worst living conditions that ended up paying the highest price in terms of both infections and deaths. This trend seems to persist over time, revealing an entrenchment of the disease in the most impoverished countries.

Another particularly relevant finding was the explanatory significance of the statistical capacity indicator, which, together with the percentage of rural population and investment in health, suggests that there has been an underestimation in the recording and registration of cases and deaths in low income countries. This has various repercussions, such as the damage caused by the pandemic going undiagnosed and therefore not being adequately addressed, or international solidarity being misdirected in the wrong direction.

One element that has made the COVID-19 pandemic unique has been the immense production and dissemination of up-to-date data on the catastrophe, on the basis of which the media discourse has been constructed and oriented towards the political management of the crisis. For this reason, this information must be monitored and disseminated with responsibility, informing about the fact that data are never an automatic translation of reality but a representation of it through measurements to which a measurement error is inexorably associated. Biased information can have serious consequences, since it will not only affect the knowledge extracted from the data themselves, but also all the scientific research developed from it. For this reason, this research recommends the use of control variables to purge biases from the statistical capacity of the country.

In summary, this research to shed light on the relationship between socioeconomic status and the spread of COVID-19. Through a comparative analysis of cross-sectional studies worldwide during two pandemic years. In addition, we included control variables never studied before, such as the statistical production capacity of the country.

The results indicate that the relationship between wealth and disease is dynamic. High income countries were more vulnerable in the early stages of the pandemic, while low income countries have been the most adversely affected in the long term. It was also found that the

statistical production capacity of each country is associated with the number of cases and deaths identified.

It is important that policymakers consider these results when designing pandemic containment plans, taking into account the special vulnerability of rich countries in the early stages of the pandemic and the greater vulnerability of impoverished countries in the long term. In addition, it is essential to keep in mind at both the policy and research levels that published disaster figures are strongly associated with each country's capacity to produce rigorous and quality statistics.

## Author Contributions

**Conceptualization:** Víctor Pérez-Segura.

**Data curation:** Víctor Pérez-Segura.

**Formal analysis:** Víctor Pérez-Segura.

**Investigation:** Víctor Pérez-Segura, Raquel Caro-Carretero, Antonio Rua.

**Methodology:** Víctor Pérez-Segura, Raquel Caro-Carretero, Antonio Rua.

**Project administration:** Raquel Caro-Carretero, Antonio Rua.

**Supervision:** Raquel Caro-Carretero, Antonio Rua.

**Validation:** Raquel Caro-Carretero, Antonio Rua.

**Visualization:** Víctor Pérez-Segura.

**Writing – original draft:** Víctor Pérez-Segura.

**Writing – review & editing:** Víctor Pérez-Segura.

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
