## [Decision Letter · Decision Letter 0]

12 Apr 2023

PONE-D-23-06829Changing Face of Socio-Economic Vulnerability and COVID-19: An Analysis of Country Wealth during the First Two Years of the PandemicPLOS ONE

Dear Dr. Pérez Segura, Thank you for submitting your manuscript to PLOS ONE. After careful consideration, we feel that it has merit but does not fully meet PLOS ONE’s publication criteria as it currently stands. Therefore, we invite you to submit a revised version of the manuscript that addresses the points raised during the review process.

We look forward to receiving your revised manuscript.

Kind regards,

George Mugambage Ruhago, PhD

Academic Editor

PLOS ONE

Journal Requirements:

Reviewers' comments:

Reviewer's Responses to Questions

**Comments to the Author**

1. Is the manuscript technically sound, and do the data support the conclusions?

Reviewer #1: Yes

Reviewer #2: Yes

2. Has the statistical analysis been performed appropriately and rigorously? 

Reviewer #1: Yes

Reviewer #2: Yes

3. Have the authors made all data underlying the findings in their manuscript fully available?

Reviewer #1: Yes

Reviewer #2: Yes

4. Is the manuscript presented in an intelligible fashion and written in standard English?

Reviewer #1: Yes

Reviewer #2: Yes

5. Review Comments to the Author

Reviewer #1: It is my pleasure to review your paper. Under the hypothesis of the dynamic and changing over time, the main objective of your paper was to determine the relationship between the country's wealth and the incidence of COVID-19. However, I have a few comments as follows;

1. There is no adequate information on the methods. The authors should indicate how data were collected, the sampling technique, the setting of your study, etc.

2. Equations 1 – 3 on pages 7 and 8, should be included in the methods instead of being presented in the results.

3. The conclusion should be summarized.

4. The authors didn’t indicate any ethical considerations. It’s important to be indicated in the study since it involved studying the incidence of COVID-19 that happened to people.

5. The list of abbreviations is missing in the main document.

Reviewer #2: The authors conducted an ecological study of global scope with a consecutive application of cross-sectional analysis during first two pandemic years of COVID 19. An analysis of country wealth was conducted. This study is important as it shows the “changing face” of the pandemic, mostly affecting high income countries at first and later the low income countries.

However, the following are my comments.

In lines 10-11…The sentence “And on the other hand to find out whether the statistical production capacity of the country is a confounding variable for COVID-19 cases and deaths” needs rephrasing.

In lines 11-12 the authors wrote that “To achieve the objectives, an ecological study of global scope is proposed”. This sentence needs rephrasing as the study has already been conducted.

In line 12-14 the sentence “Whose analytical strategy is the consecutive application of cross-sectional analysis…” should be rephrased.

The authors conclude that poor statistical production capacity of poorer countries may be masking the real incidence of the disease. However, the definition of poor statistical production and was not clearly explained. In addition, I would suggest the use of “low income countries” rather than “poorer countries” in the statement.

6. PLOS authors have the option to publish the peer review history of their article (what does this mean?). If published, this will include your full peer review and any attached files.

Reviewer #1: **Yes: **Malale Tungu

Reviewer #2: No

---

## [Author Response · Author response to Decision Letter 0]

31 May 2023

Dear Professors,

We would like to thank the reviewers to help us to improve the quality of this manuscript with their helpful comments and constructive suggestions, we have introduced several clarifications and comments in the manuscript that we consider that can help to their better understanding.

We look forward to your response,

The authors

Response. Reviewer 1

1. There is no adequate information on the methods. The authors should indicate how data were collected, the sampling technique, the setting of your study, etc.

- In the methods section, a sub-section called measures has been included where information on the units of analysis and measures has been included (Pages 6-10).

2. Equations 1 – 3 on pages 7 and 8, should be included in the methods instead of being presented in the results.

- The equations have been included in the methods section (Page 6) along with a brief explanation of the analytical strategy (Page 5, lines 20-25; Page 6, lines 1-2).

3. The authors didn’t indicate any ethical considerations. It’s important to be indicated in the study since it involved studying the incidence of COVID-19 that happened to people. 

- Since this is a macro analysis at the national level on data from secondary data sources, the work does not jeopardize the privacy of the participants. All data used in this analysis are publicly available and can be found at: : https://github.com/vicperez/Changing-Face-of-Socio-Economic-Vulnerability-and-COVID-19 (Page 6; lines 26-30).

4. The conclusion should be summarized. 

- A summary of the paper's findings and contributions is included at the end of the article (Page 16, lines 7-21).

5. The list of abbreviations is missing in the main document. 

- A list of abbreviations has been included at the beginning of the main manuscript (page 2, line 6).

Response. Reviewer 2

In lines 10-11…The sentence “And on the other hand to find out whether the statistical production capacity of the country is a confounding variable for COVID-19 cases and deaths” needs rephrasing

The sentence has been rephrased as: “And on the other hand, to find out if the country's statistical production capacity is masking the real incidence of the COVID-19 pandemic” (Page 2, lines 13-14).

In lines 11-12 the authors wrote that “To achieve the objectives, an ecological study of global scope is proposed”. This sentence needs rephrasing as the study has already been conducted.

The sentence has been rephrased as: In order to achieve this objective, an ecological study has been designed at international level with the countries established as study units” (Page 2, lines 14-16)

In line 12-14 the sentence “Whose analytical strategy is the consecutive application of crosssectional analysis…” should be rephrased.

The sentence has been rephrased as: “The analytical strategy utilized involves the consecutive application of cross-sectional analysis, specifically employing multivariate linear regression on a daily basis throughout the first two years of the pandemic (from 03/14/2020 to 03/28/2022)” (Page 2, lines 16-18).

The authors conclude that poor statistical production capacity of poorer countries may be masking the real incidence of the disease. However, the definition of poor statistical production and was not clearly explained. In addition, I would suggest the use of “low income countries” rather than “poorer countries” in the statement

- Statistical production capacity has been defined in the subsection Measures of the section Method (Pages 6-10)

- Poor countries have been replaced by low income countries (Page 3, line 5; Page 15, line 19; Page 16, lines 12-13),

---

## [Decision Letter · Decision Letter 1]

11 Aug 2023

Changing Face of Socio-Economic Vulnerability and COVID-19: An Analysis of Country Wealth during the First Two Years of the Pandemic

PONE-D-23-06829R1

Dear Dr. Pérez Segura,

We’re pleased to inform you that your manuscript has been judged scientifically suitable for publication and will be formally accepted for publication once it meets all outstanding technical requirements.

Kind regards,

Ricardo de Mattos Russo Rafael, Ph.D.

Academic Editor

PLOS ONE

Reviewers' comments:

Reviewer's Responses to Questions

**Comments to the Author**

1. If the authors have adequately addressed your comments raised in a previous round of review and you feel that this manuscript is now acceptable for publication, you may indicate that here to bypass the “Comments to the Author” section, enter your conflict of interest statement in the “Confidential to Editor” section, and submit your "Accept" recommendation.

Reviewer #1: All comments have been addressed

Reviewer #2: All comments have been addressed

2. Is the manuscript technically sound, and do the data support the conclusions?

Reviewer #1: Yes

Reviewer #2: Yes

3. Has the statistical analysis been performed appropriately and rigorously? 

Reviewer #1: Yes

Reviewer #2: Yes

4. Have the authors made all data underlying the findings in their manuscript fully available?

Reviewer #1: Yes

Reviewer #2: Yes

5. Is the manuscript presented in an intelligible fashion and written in standard English?

Reviewer #1: Yes

Reviewer #2: (No Response)

6. Review Comments to the Author

Reviewer #1: (No Response)

Reviewer #2: The authors have adequately addressed the comments raised previously and the manuscript is technically sound.

7. PLOS authors have the option to publish the peer review history of their article (what does this mean?). If published, this will include your full peer review and any attached files.

Reviewer #1: No

Reviewer #2: No

---

## [Editor Report · Acceptance letter]

18 Aug 2023

PONE-D-23-06829R1 

Changing Face of Socio-Economic Vulnerability and COVID-19: An Analysis of Country Wealth during the First Two Years of the Pandemic 

Dear Dr. Pérez-Segura:

I'm pleased to inform you that your manuscript has been deemed suitable for publication in PLOS ONE. Congratulations! Your manuscript is now with our production department. 

Kind regards, 

on behalf of

Dr. Ricardo de Mattos Russo Rafael 

Academic Editor

PLOS ONE